# Numerical Study on the Effect of Coarse Aggregate Shape during Concrete Mixing Process

**DOI:** 10.3390/ma17071515

**Published:** 2024-03-27

**Authors:** Jianjun Shen, Binqiang Wang, Jingru Hou, Pengchao Yao

**Affiliations:** 1Key Laboratory of Road Construction Technology and Equipment, Ministry of Education, Chang’an University, Xi’an 710064, China; sjjun7406@sina.com (J.S.); wbq874283567@163.com (B.W.); 2CCCC Xi’an Road Construction Machinery Co., Ltd., Xi’an 712000, China; yaopc1993@163.com

**Keywords:** coarse aggregate shape, concrete mixing, discrete element, mixing mechanism

## Abstract

The shape of coarse aggregate is an important factor determining the performance of concrete, and it also affects the whole mixing process. This article selected four typical coarse aggregates and simulated the motion behavior of the coarse aggregate and mortar in a dual horizontal axis mixer using discrete element software, EDEM. The mixing motion of coarse aggregates with different shapes and mortar was studied using average velocity, contact rate, and dispersion coefficient as indicators. The results show that the largest average speed in the mixing process was achieved with the spherical coarse aggregate. Flat coarse aggregates have the highest velocity in the vertical direction, while ellipsoidal coarse aggregates have the lowest velocity. The spherical coarse aggregate mixes best with mortar while the ellipsoidal coarse aggregate mixes worst with mortar. The results of the study could provide strategies for the concrete mixing process considering the shape of the aggregate.

## 1. Introduction

Concrete is the largest material and part used in building construction [1], and its quality has a significant impact on the quality of building construction [2], such as bridges, airports, roads, and military defenses. Concrete is usually made of coarse aggregate, gravel [3,4], cement, and other materials in appropriate proportions, relying on a horizontal shaft mixing device and the selection of appropriate mixing parameters [5,6,7] to complete its mixing production process. Therefore, the quality of concrete often depends on various factors such as device structural parameters, mixing parameters, and material characteristics during the mixing process [8,9,10]. In terms of structural parameters, researchers mainly use orthogonal experimental methods to analyze the uniformity of the mixed materials, aiming to determine the optimal matching parameters for the installation angle of the mixing device, such as the blade inclination angle and mixing arm phase angle [11,12,13,14]. In terms of mixing parameters, the influence of changes in parameters such as mixing shaft speed, filling level, and material feeding order on material mixing performance can be studied to improve mixing efficiency. In terms of material characteristics, the influence of material particle characteristics on mixing performance and particle flow characteristics has been analyzed accordingly [15,16].

When studying the concrete mixing process, due to the large volume and opacity of the mixing device, it can be difficult to observe the particle state of the material, evaluate the impact of various factors on the quality of concrete, and the experimental cost is expensive [17,18]. Therefore, EDEM software was introduced in the numerical simulation research of concrete mixing. The software uses the discrete element method (DEM) to simulate the advantages of granular behavior of granular materials through discontinuous modeling [19,20,21], simulate fresh concrete as granular fluid, coarse aggregate particles, and mortar particles [22,23], and then carry out factor analysis, parameter optimization and mixing mechanism research on the concrete mixing process [24,25,26].

The morphological characteristics of coarse aggregates have a significant Impact on the workability of concrete. Figure 1 shows the flow state of different forms of coarse aggregates in fresh concrete [27]. Coarse aggregate particles mainly play a skeleton-supporting role in concrete, and their shape is one of the important parameters that must be considered in the numerical simulation modeling of concrete mixing based on EDEM. However, EDEM defaults to using spherical particles as the basic particle model, and much of the literature directly selects spherical particles to simulate coarse aggregate particles to simplify the modeling process [28,29]. This differs greatly from the actual shape of coarse aggregate particles, making it difficult to truly simulate the concrete mixing process. Therefore, some scholars have proposed nonspherical particle modeling methods based on randomly shaped polyhedral [30] and image segmentation methods [31], which can generate arbitrarily shaped clumps through overlapping spherical algorithms. Although the above methods can establish a real coarse aggregate particle model for numerical simulation research, each batch of coarse aggregate has different shapes in large numbers during the experiment. If the coarse aggregate particle model needs to be reestablished every time the simulation is conducted, the process is not only cumbersome, but also requires a large amount of computational resources.

To address the above problems, this article establishes a numerical simulation method for the influence of coarse aggregate shape on the concrete mixing process. Firstly, approximate models of four representative coarse aggregate models were established in 3D modeling software, and then the models were imported into EDEM software for automatic particle filling to obtain coarse aggregate particle models of different shapes. Then, relying on a dual horizontal axis mixing device and combining different shapes of coarse aggregate models, the influence of coarse aggregate particle motion routes, dispersion coefficients, contact rates, and other indicators on the concrete mixing process are studied. The results of this study could provide strategies for the concrete mixing process considering the shape of the aggregate.

## 2. Model and Discrete Element Method

### 2.1. Model of Mixer

The dual horizontal shaft forced mixer is currently the main type of mixing equipment, consisting of mechanisms such as transmission, mixing, unloading, shaft end sealing, and support [32,33]. The mixing mechanism is the core part of the dual horizontal shaft mixer, which determines the quality of concrete mixing. This article mainly studies the characteristics of coarse aggregate particles, therefore simplifying the mixer. The simplified mixer is shown in Figure 2, consisting of a mixing drum, mixing shaft, and mixing blades, and its values are shown in Table 1.

The number of blades installed on a single mixing shaft was 7, and the phase angle for the installation of adjacent blades was 90°. The blade installation angle α was 37°. The mixing blades were arranged in a staggered manner to avoid interference, with a forward and reverse arrangement. To facilitate simulation calculations and reduce computational complexity, this part of the model only retained the mixing drum and mixing device, while other devices such as transmission devices and unloading devices were not considered. The mixer was modeled in SolidWorks and imported into EDEM. Through consulting literature materials, the optimal speed for blade stirring is a linear speed of 1.5 m/s [34], which is converted into a stirring shaft speed of 77 rpm. The two shafts had the same speed but in opposite directions.

### 2.2. Model of Coarse Aggregate

According to the provisions, the coarse aggregates for cement concrete can be categorized as crushed stone and pebbles. When the same concrete mix ratio is used, concrete mixed with pebbles has better fluidity but lower strength after hardening, while concrete mixed with crushed stone has poorer fluidity but higher strength after hardening [35]. Coarse aggregate shapes are usually categorized into four categories: flat, triangular cone, ellipsoidal, and spherical [36]. The first two shapes are the most common crushed stone shapes, while the last two shapes are pebble shapes. When modeling the shape of the coarse aggregate, a three-dimensional model was first established in the modeling software SolidWorks (version 2021) based on the actual shape of the coarse aggregate and then imported into the EDEM software. The EDEM software comes with a particle filling function, which fills the 3D model with the most suitable number and size of particles. Finally, a complete coarse aggregate model was obtained, and the modeling process of coarse aggregate particles is shown in Figure 3.

According to the reference of coarse aggregate particle size distribution provided by the industry standard [37], this article generated coarse aggregate particles in EDEM with a particle size range of 5–40 mm using a random generation method. When modeling the mixing process, only cementitious materials, fine sand, and coarse aggregate models were considered. The influence of water on the mixing process and the cohesive effect of cementitious materials was characterized by the setting surface energy parameters. Due to the fusion of cementitious materials, fine sand, and water to form mortar, fresh concrete was considered as a mixture of coarse aggregate and mortar during modeling. This paper referred to the modeling process of other scholars [38,39,40] to set the particle radius of the mortar to 3 mm in all cases.

### 2.3. Discrete Element Method Theory and Contact Models

Since concrete consists of coarse aggregate particles, mortar particles, etc., and there is a strong adhesion between the two particles, the effect of adhesion between the particles must be considered. To characterize the viscosity between particles, the contact model between particles was selected as the *JKR* (Hertz–Mindlin with Johnson Kendall Roberts) contact model. The traditional Hertz model is a special case of when the surface energy of the *JKR* contact model is zero [41]. When using EDEM for discrete element analysis, the normal contact force between model particles is calculated by *F_JKR_*:(1)$$F_{JKR}=-4\sqrt{\pi \gamma E^{*}}\alpha^{\frac{3}{2}}+\frac{4E^{*}}{3R^{*}}\alpha^{3}$$

In Equation (1), *F_JKR_* is the *JKR* normal force (*N*),
γ is the surface energy *J*·m^−2^, $E^{*}$ is the equivalent Young’s modulus, $R^{*}$ is the equivalent radius, and α is the tangential overlap.

The relationship between tangential overlap (α) and normal overlap (δ) is as follows:(2)$$\delta =\frac{\alpha^{2}}{R^{*}}-\sqrt{4\pi \gamma \alpha /E^{*}}$$

In Equation (2), the equivalent Young’s modulus $E^{*}$ and the equivalent radius $R^{*}$ are defined in Equations (3) and (4):(3)$$\frac{1}{E^{*}}=\frac{\left(1-\upsilon_{i}^{2}\right)}{E_{i}}+\frac{\left(1-\upsilon_{i}^{2}\right)}{E_{j}}$$
(4)$$\frac{1}{R^{*}}=\frac{1}{R_{i}}+\frac{1}{R_{j}}$$

In Equations (3) and (4), $E_{i}$, $\upsilon_{i}$, $R_{i}$ and $E_{j}$, $\upsilon_{j}$, $R_{j}$ are the Young’s modulus, Poisson’s ratio, and the radius of the contact sphere, respectively.

To set the model pre-processing contact parameters, this article conducted slump experiments for parameter calibration [42], and set the contact parameters of various materials in the simulation environment to fit the flowability of real concrete. Figure 4a–c shows the diagram of the experiment slump. Experiments were conducted with slump test tubes to obtain the corresponding concrete slump and expansion values. Figure 4d–f shows the simulation of a slump. A slump cylinder, mortar, and coarse aggregate model was established in the simulation environment to simulate the slump experiment. After repeated simulations, simulation parameters close to the real situation were obtained as shown in Table 2. The mixer was made of steel. The Poisson’s ratio, Shear modulus, and density of sand, stone, and cement particles are shown in Table 3.

## 3. Numeric Simulation Results and Analysis

In the DEM research process, a concrete mixing model was established according to Section 2, and the research process is shown in Figure 5. The mixer and coarse aggregate were modeled in SolidWorks software and imported into EDEM software for numerical simulation analysis of concrete mixing. Using the shape of the coarse aggregate as the experimental factor, and collision rate, average velocity, and dispersion coefficient as the experimental indicators, the influence of different coarse aggregate shapes on concrete mixing was studied.

For numerical simulations, mortar particles were first generated at the top of the mixer and placed at the bottom of the mixer, followed by coarse aggregate particles. The particles all fall into the mixing drum at a speed of 1 m/s. The number of mortar particles was 60,000, and the number of coarse aggregate particles was 2000, which was generated within 1 s. The mixer started to rotate at 1 s, the rotation stopped at 10 s and the mixing was finished. The mixing of coarse aggregates with different shapes and mortar particles is shown in Figure 6. From Figure 6, it can be seen that the coarse aggregate and mortar have been thoroughly mixed.

As shown in Figure 7, representative cross-sections A and B were taken, and the characteristics of the velocity vector, particle distribution, and other characteristics of the material on the representative cross-sections were analyzed to describe in detail the mixing and dispersion motion of the material [43].

### 3.1. The Influence of Coarse Aggregate Shape on Concrete Mixing

#### 3.1.1. Movement of Coarse Aggregate

The concrete raw material was dropped from above and under the action of the two mixing shafts made a circular motion from both sides to the center. The particles collide extensively in the center region, thus mixing. After passing through the center region, the material moves towards both sides of the mixing drum under the action of stirring force and then returns to the center with the stirring blades. The material is mixed evenly in a cyclic manner.

To understand the movement of particles during the mixing process, this article analyzes the velocity distribution state of particles in typical cross sections. The particle velocity distribution in section A at a certain time in the mixing process was selected.

Figure 8 shows the particle velocity vector diagram at a certain moment in section A. It can be seen that the particle with the highest velocity is located near the mixing axis rotating downwards on both sides. Under the dual effects of the stirring force of the stirring shaft and its gravity, the particles near the rotary mixing shaft close to the cylinder on both sides move most intensely. Most particles with average speed are concentrated in the middle part of the dual horizontal shaft mixer. The particles in the middle part are subjected to the stirring force of the stirring shaft, overcoming the force of gravity and moving upwards. Due to the opposite mixing directions of the mixing shafts on both sides, a large number of collisions occur between the particles on both sides in the middle, resulting in a decrease in particle velocity due to collision compression and self-gravity. The particles with the smallest speed in the dual horizontal axis mixer are located at the bottom of the mixing drums on both sides. The bottom particles are squeezed by the upper particles, while the force of the stirring shaft cannot be transmitted to the bottom layer, resulting in a very low velocity of the bottom particles.

Figure 9 shows the average speed of coarse aggregates of various shapes during the mixing process, with spherical coarse aggregates having the highest average speed and ellipsoidal coarse aggregates having the lowest speed. Because the contact point and contact area of spherical coarse aggregates do not change significantly over time, they are not subjected to sudden stresses in different directions, resulting in the highest average velocity. Ellipsoidal coarse aggregates have a small stress area in the middle due to their long sides and short middle and are prone to asymmetric forces on both sides. Therefore, it is difficult for the mixing force to drive the movement of ellipsoidal coarse aggregate, resulting in the minimum average speed. The average speed of the flat coarse aggregate is higher than that of the triangular cone coarse aggregate, indicating that during the mixing process the flat coarse aggregate moves more fiercely than the triangular cone coarse aggregate and more easily overcomes the mutual impact and gravity effect.

The particle mixing process mainly depends on the convective motion of the particles under the impact of the rotating stirring shaft and the self-falling motion of their gravity. To ensure that particles at each position in the mixer are subjected to stirring, coarse aggregate particles from three different locations C, D, and E, as shown in Figure 10, were selected as marker particles. By analyzing the trajectory of the marker particles in the vertical direction, the stirring effect they are subjected to is analyzed.

Figure 11a–c represents the motion trajectories of the labeled coarse aggregate particles C, D, and E at different positions in the vertical direction. From Figure 11a–c, it can be inferred that under the action of two rotating mixing shafts, each coarse aggregate moves up and down in the influence area of the mixing shaft. The movement of four different shapes of coarse aggregates at three different positions is similar, with all following the movement of the mixing shaft. This indicates that the coarse aggregate in a dual horizontal shaft mixer can move under the action of the mixing shaft regardless of its shape and position.

Figure 11d shows the average vertical velocity at three positions of coarse aggregates of different shapes. The average vertical velocities of flat, triangular cone, ellipsoidal, and spherical coarse aggregates are 0.2172 m/s, 0.1993 m/s, 0.1824 m/s, and 0.1997 m/s, respectively. The results indicate that the flat coarse aggregate has the fastest speed in the vertical direction and is more likely to overcome gravity and work under the action of the mixing axis. The movement speed of triangular cone, and spherical coarse aggregates in the vertical direction is similar. The ellipsoidal coarse aggregate has the smallest velocity in the vertical direction and is imbalanced in force, making it difficult to move in the vertical direction.

#### 3.1.2. Collision of Coarse Aggregate

During the concrete mixing process, with the rotation of the mixing shaft, the coarse aggregate and mortar collide at all times. To understand the collision between different shapes of coarse aggregates and mortar during the mixing process, section A was selected as the observation surface to observe the material distribution of four different shapes of coarse aggregates at three different times.

Figure 12 shows the distribution of coarse aggregate and mortar at different times in section A, and it can be seen from a macro perspective that all shapes of coarse aggregate are fully mixed with the mortar. During the mixing process, a large number of collisions occur between the coarse aggregate and the mortar, mixing shaft, cylinder, etc. During the continuous collision process, the coarse aggregate and mortar are mixed more evenly. Most of the collision of materials in the mixer occurs in the middle area. Since the two mixing shafts turn in opposite directions, there is an overlapping area between the two mixing shafts in the middle part where particles will have a lot of impact and collision.

To gain a detailed understanding of the motion and collision of materials in the mixer, section B was selected as the observation surface. This was to study the motion and collision of materials in the central area of the mixing shaft, which is the place where the collision is most intense. Figure 13a shows the main view and top view of the cross-section of the middle area taken, with the green box indicating the middle area taken. In this area, there is an overlapping mixing area between the two mixing shafts. Due to the opposite rotation of the two mixing shafts, both of them rotate upwards in the middle area. The mixing blades have a certain angle, allowing the material to move both up and back. Figure 13b shows the velocity cloud map of the material in section B. The upper two blades are on the left mixing shaft, and the lower two blades are on the right mixing shaft. When the upper blade rotates upwards, it moves the animal material up and right. Correspondingly, the lower blades move up and left with animal material. Therefore, while stirring and mixing, the mixing blades carry the material in a circular motion in the mixing drum, moving back and forth.

After analyzing the collision of materials during the mixing process on macro and fine scales, it is also necessary to view numerically the collision of different shapes of coarse aggregates with the mortar. In this article, the collision number rate was selected to characterize the collision situation between the coarse aggregate and mortar, and its formula is as follows [44]:(5)$$q=c_{i}/c$$

In Equation (5), $c_{i}$ is the number of contacts between different particles, and c is the number of contacts between all particles. The higher the q, the more intense the collision.

Figure 14 shows the collision rate of coarse aggregates with different shapes. It can be seen that the collision rate of spherical coarse aggregates is higher than that of the other three shapes of coarse aggregates. During the mixing process, the collision of spherical coarse aggregates is the most intense and the movement rate is also the highest. The collision rate of the triangular cone-shaped coarse aggregate is the lowest. The collision rate of the ellipsoidal shape is similar to that of the flat coarse aggregate.

### 3.2. The Influence of Coarse Aggregate Shape on Concrete Mixing

Mixing uniformity is an important indicator to measure the mixing quality of the mixing drum. To accurately quantitatively analyze the changes in particle uniformity during the mixing process, the mixing drum model is divided into the same grid group, as shown in Figure 15. The number of grid divisions was 500, and the grid division effect is shown in Figure 15. Then, the two types of particle numbers and the total number of particles in each grid were recorded at time intervals of 0.5 s. To ensure a sufficient number of particles in the calculated grid and reduce random errors caused by a small number of particles in the grid, grids with a total number of particles less than 100 are not considered in the actual calculation process.

There are various mixing indicators used to characterize mixing performance, such as the Lacey mixing index, segregation index, generalized average mixing index, etc. To evaluate the mixing performance of the mixer, this article uses the mathematical statistical method of discrete coefficient (*L*) to quantify the mixing state of particles in the mixer [45]. The larger the dispersion coefficient, the more uneven the stirring effect, while the opposite indicates a good stirring effect.
(6)$$\varepsilon =\frac{n_{i}/n}{N_{i}/N}(i=1,2,\ldots )$$
(7)$$L=\sqrt{\frac{\sum_{i=1}^{j}{\left(\varepsilon_{i}-\varepsilon_{avg}\right)}^{2}}{j-1}}$$

In Equations (6) and (7), $n_{i}$ and n represents the number of particles in the grid and the total number of particles in the grid, respectively, $N_{i}$ and N represents the number of particles in the mixer and the total number of particles, ε represents the deviation degree of the actual and ideal particle uniformity in the grid, $\varepsilon_{i}$ represents the deviation degree of particles in the grid, $\varepsilon_{avg}$ represents the average deviation degree, and j represents the number of grids.

Figure 16 divides the mixing process into three stages (I, II, III) and analyzes the possible reasons for the formation of this discrete coefficient curve. In stage I, the mixer has just started, and the mixing shaft and blades begin to rotate. Due to the layered addition of mortar and coarse aggregate into the mixer, the contact surface between the two materials only has a transition layer in the middle, resulting in a large dispersion coefficient at the beginning. Under the action of mixing, the mortar and coarse aggregate begin to come into contact and collide, and the dispersion coefficient begins to rapidly decrease. In stage II, the motion of particles gradually forms shear flow and convection. The uniformity of particle mixing gradually decreases slowly, mainly relying on the mutual penetration and diffusion between particles. In stage III, the coarse aggregate and mortar have been mixed evenly, and the fluctuation of the dispersion coefficient is not significant.

As shown in Figure 16, the dispersion coefficient between coarse aggregates of different shapes and mortar gradually decreases over time under the action of mixing. The dispersion coefficient curve of the spherical coarse aggregate and mortar changes the fastest, and the dispersion coefficient value is the smallest, indicating that the mixing effect of spherical coarse aggregate and mortar is the best. The dispersion coefficient between the ellipsoidal coarse aggregate and mortar has always been the highest among the four types of coarse aggregate, and its mixing effect with mortar is the worst. The dispersion coefficient curves of the flat and triangular cone-shaped coarse aggregates differ in the early stages but are similar in the middle and later stages. Therefore, the mixing situation of these two types of coarse aggregates and mortar is similar.

## 4. Conclusions

This article uses the EDEM discrete element simulation software to simulate the mixing effect of coarse aggregate and mortar in a dual horizontal axis mixer. The mixing of four different shapes of coarse aggregate and mortar was investigated, and the following conclusions were obtained:(1)The spherical coarse aggregate has the highest average speed, while the ellipsoidal coarse aggregate has the lowest average speed. The movement of the spherical coarse aggregate is the most intense in the mixing process, and it is easier to overcome the mutual impact of this coarse aggregate and its gravity.(2)The flat coarse aggregate has the fastest speed in the vertical direction, and the triangular cone-shaped and spherical coarse aggregate has a similar speed in the vertical direction. The ellipsoidal coarse aggregate has the smallest speed in the vertical direction, the force is unbalanced, and it is difficult to move in the vertical direction.(3)The spherical coarse aggregate with mortar has the fastest change in the curve of the coefficient of dispersion, the smallest value of the coefficient of dispersion, and the best mixing effect with mortar. The ellipsoidal coarse aggregate has the worst mixing effect with mortar. The discrete coefficient curves of the lamellar and triangular cone coarse aggregates are different in the early stages and similar in the middle and late stages, and the mixing of these two kinds of coarse aggregates with mortar is similar.(4)In the process of concrete mixing, the spherical coarse aggregate is the easiest to mix evenly, but for the ellipsoidal coarse aggregate the mixing time should be prolonged and the mixing intensity increased.

Since this paper is based on numerical simulation of the movement of coarse aggregates with different shapes in a mixer, it is impossible to simulate the physical and chemical reactions during the mixing operation. Therefore, follow-up research can be guided by the simulation results and verified by the actual experimental results.

## Figures and Tables

**Figure 1 materials-17-01515-f001:**
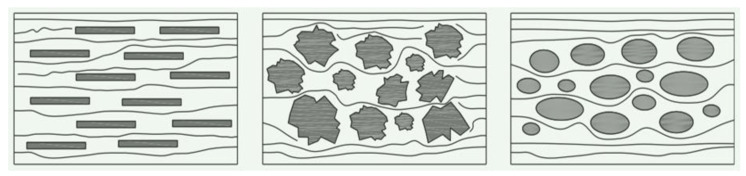
Schematic illustration of the flow state of coarse aggregate with different morphologies in fresh concrete [27].

**Figure 2 materials-17-01515-f002:**
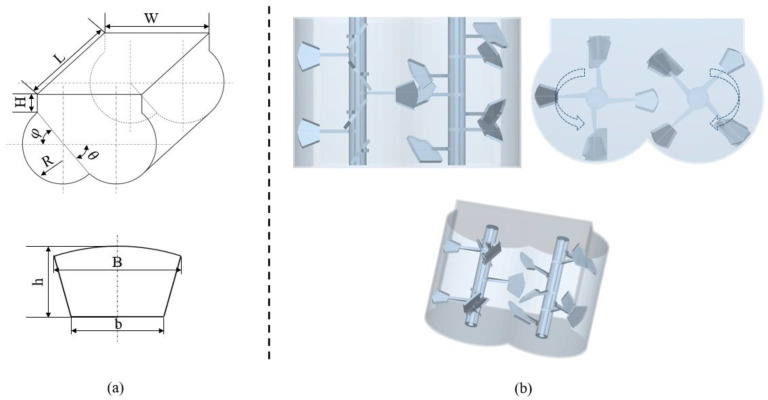
Model of a dual horizontal shaft mixer: (**a**) structure and blade size of the dual horizontal shaft mixer; (**b**) modeling of double horizontal axis mixer.

**Figure 3 materials-17-01515-f003:**
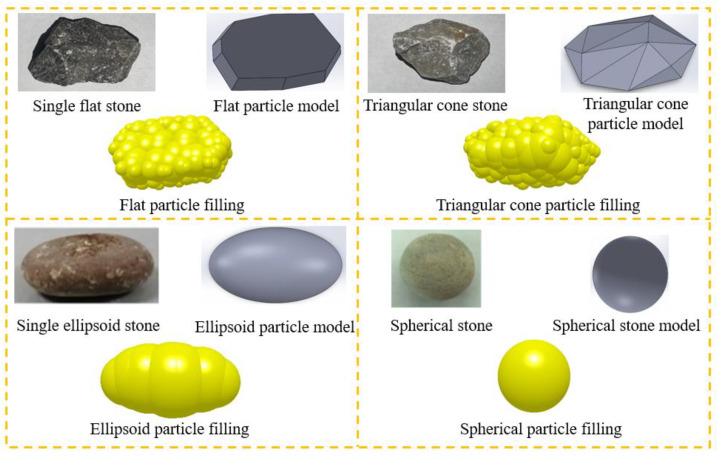
Modeling of particles with different coarse aggregate shapes.

**Figure 4 materials-17-01515-f004:**
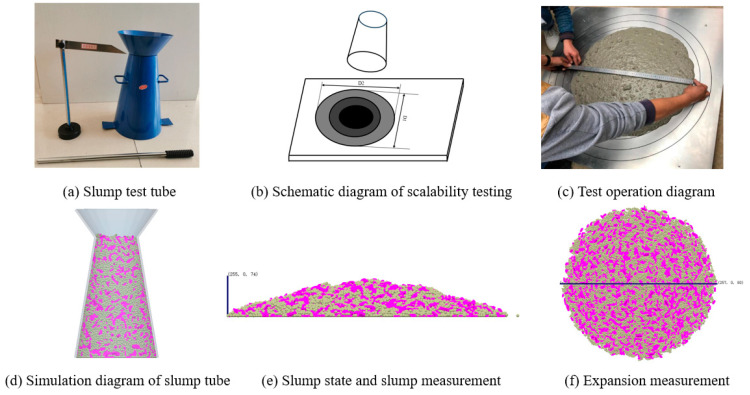
Contact parameter calibration test and simulation: (**a**–**c**) slump experiment; (**d**–**f**) slump simulation.

**Figure 5 materials-17-01515-f005:**
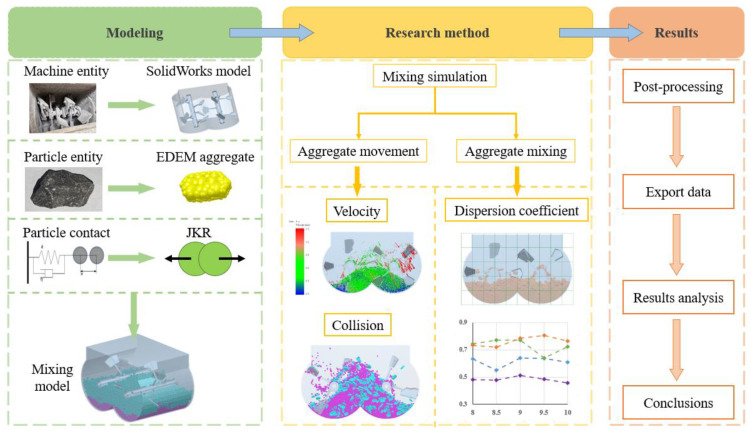
Research flowchart.

**Figure 6 materials-17-01515-f006:**
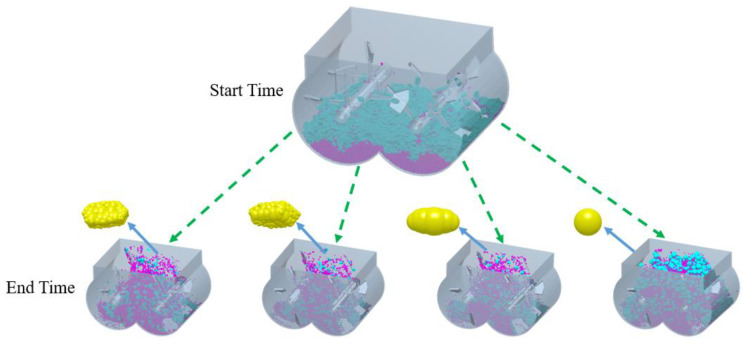
Mixing state diagram of different coarse aggregates.

**Figure 7 materials-17-01515-f007:**
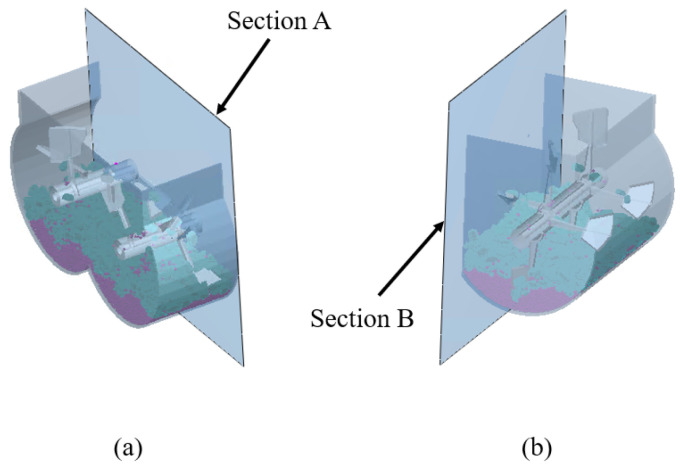
Schematic diagram of cross-section positioning for dual horizontal shaft mixer: (**a**) Section A; (**b**) Section B.

**Figure 8 materials-17-01515-f008:**
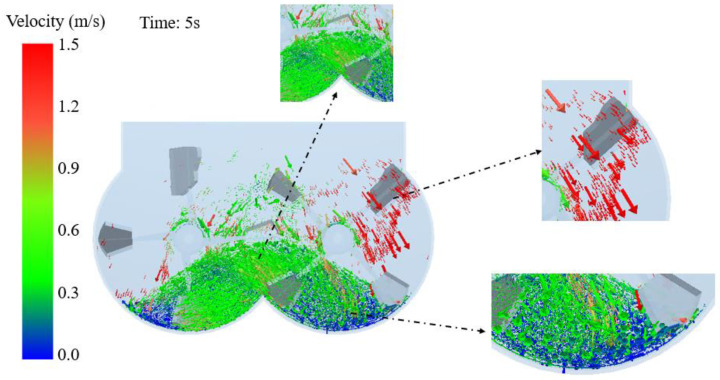
Particle velocity vector diagram at a certain moment in section A.

**Figure 9 materials-17-01515-f009:**
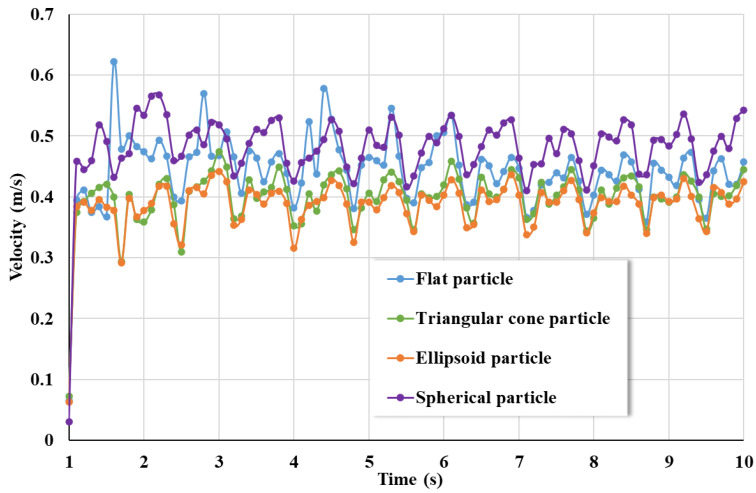
The average speed of coarse aggregates of various shapes during the mixing process.

**Figure 10 materials-17-01515-f010:**
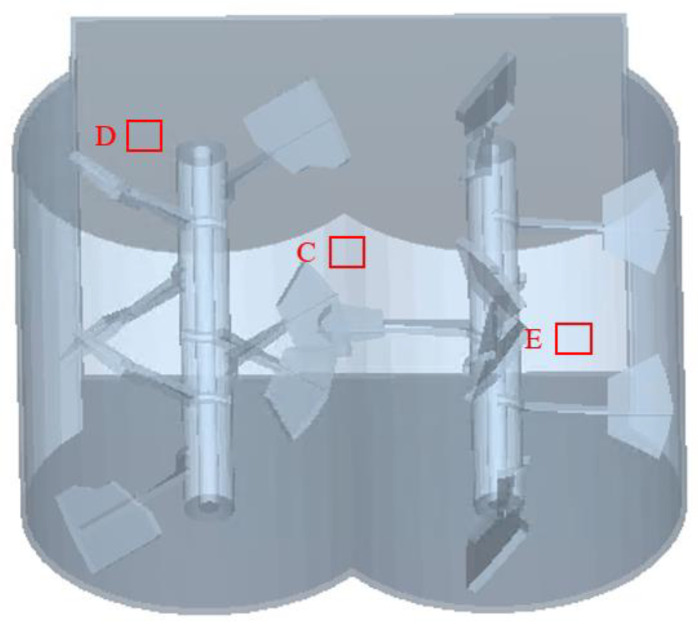
Mark the position of particles.

**Figure 11 materials-17-01515-f011:**
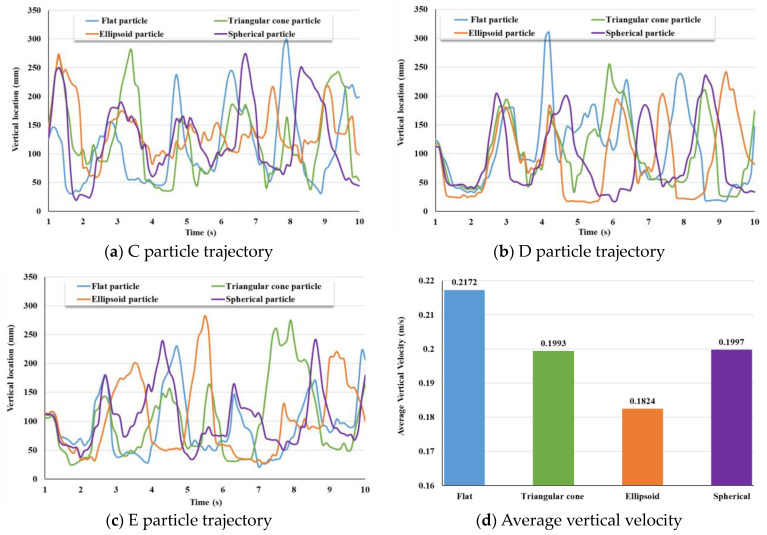
(**a**–**c**) show vertical trajectories of labeled coarse aggregates of different shapes at different positions; (**d**) average vertical velocity of coarse aggregates of different shapes at different positions.

**Figure 12 materials-17-01515-f012:**
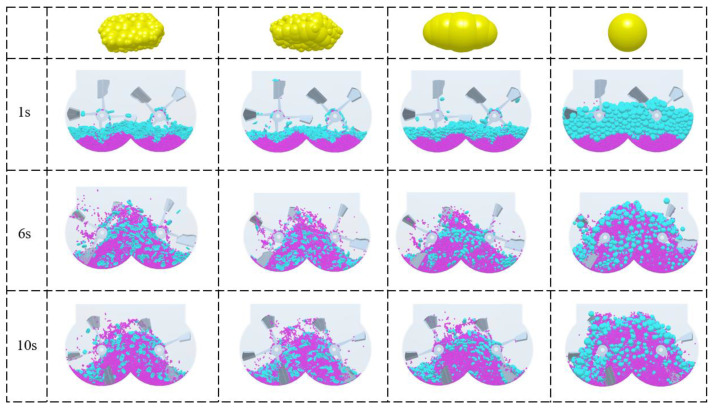
Distribution diagram of mixed coarse aggregates and mortar of various shapes in section A of the mixer.

**Figure 13 materials-17-01515-f013:**
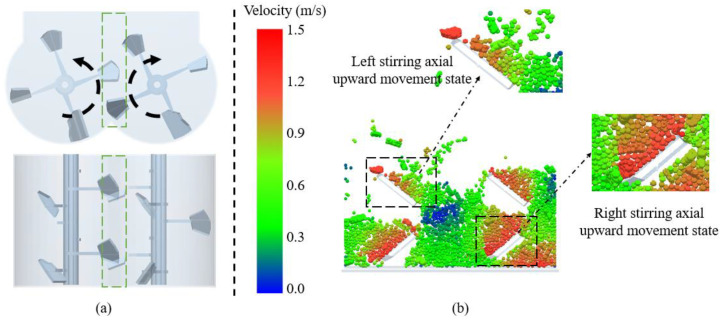
The velocity cloud map of the material in the middle area: (**a**) the main and top views of the middle area; (**b**) the velocity cloud map of the material in section B.

**Figure 14 materials-17-01515-f014:**
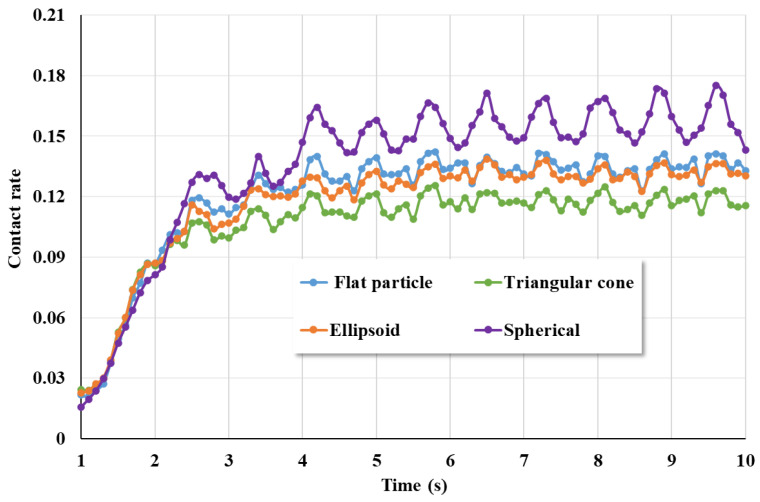
Collision rate of coarse aggregates with different shapes.

**Figure 15 materials-17-01515-f015:**
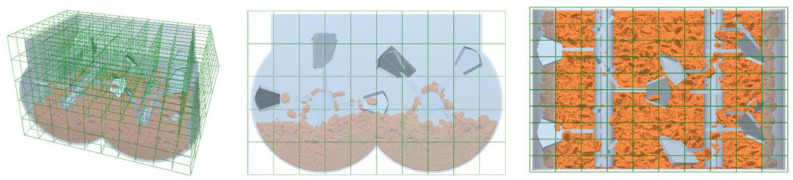
Divide the grid.

**Figure 16 materials-17-01515-f016:**
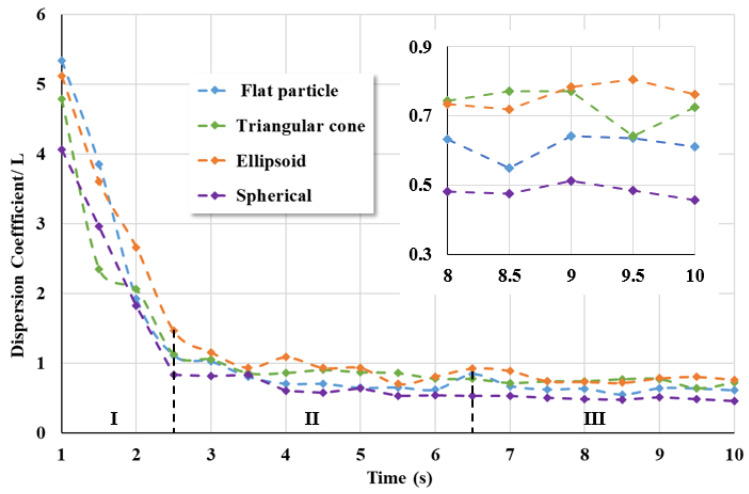
The variation of dispersion coefficient between coarse aggregates of different shapes and mortar over time.

**Table 1 materials-17-01515-t001:** Size of mixing drum and mixing blade.

Parameter (Unit)	Value
Cylinder radius, R (mm)	186
Cylinder length, L (mm)	433
Cylinder width, W (mm)	548
Difference between overall height and cylinder radius, H (mm)	93
Angle 1, θ (°)	40
Angle 2, φ (°)	45
Height of blade, h (mm)	65
The long axis of the blade, B (mm)	92
Short shaft of the blade, b (mm)	54

**Table 2 materials-17-01515-t002:** Discrete element method for simulating input parameters of particle systems.

Contact Pairs	Coefficient of Restitution	Coefficient of Static Friction	Coefficient of Rolling Friction	Surface Energy
Aggregates–Aggregates	0.15	0.07	0.05	1.2
Aggregate–Mortar	0.05	0.08	0.06	3
Aggregates–Steel	0.2	0.05	0.03	0.8
Mortar–Mortar	0.01	0.16	0.12	8
Mortar–Steel	0.03	0.11	0.09	1

**Table 3 materials-17-01515-t003:** Physical parameters of materials.

Category	Density (kg/m^3^)	Shear Modulus /(GPa)	Poisson’s Ratio
Aggregate	2600	20	0.35
Mortar	2100	2	0.25
Steel	7800	70	0.3

## Data Availability

Data are contained within the article.

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
