# Peer review of "Numerical Study on the Effect of Coarse Aggregate Shape during Concrete Mixing Process"

_materials, 2024, doi:10.3390/ma17071515_

Round 1

Reviewer 1 Report

Comments and Suggestions for Authors

The manuscript presents a numerical simulation of the concrete mixing process. The authors used the Discrete Element Method and EDEM software. The study included the influence of the shape of aggregate particles on their movement in the mixture and mixing time. The research is clearly described and the conclusions are correctly formulated.

Methods for examining the movement of grains in a fresh concrete mixture based on the Discrete Elements Method have been intensively developed in the last decade. The article fits into this trend, which proves its relevance. Unfortunately, as in the case of most studies of this type, there is no laboratory confirmation of the obtained results of numerical simulation. Of course, I understand that examining the movement of a single grain in the mixture is rather impossible, but, for example, a rough examination of the quality of mixing with a visual assessment is easy to implement.

The method used is at the development stage, so I assume it will not be verified, but it is worth considering by the authors in the future.

I have a few comments that require clarification in my opinion:

1) Does the mixer model shown in Figure 1 correspond to any actual device available on the market?

2) What do the authors understand by the term "continuous particle size grading" (line 112) - the same number of aggregate grains for different size ranges?

3) Isn't the diameter of 3 mm on the mortar particles too large? Sand grain is max. 2mm, much less cement.

4) aggregate grain sizes are overscaled similarly (40mm 111 - in practice we rarely use particles larger than 16mm)

5) line 330, I don't understand the difference between "ni" (number of particles in the grid) and "n" (total number of particles in the grid), shouldn't there be "number of aggregate particles" for ni? A similar comment applies to "Ni" and "N"

6) Figure 13 - the first mixing phase allows obtaining an almost homogeneous mixture, in phases II and III the L coefficient changes only slightly. It's surprising to me that just 2.5 seconds is enough to achieve such quality mixing. Do the authors know of any studies that would confirm this result?

Reviewer 2 Report

Comments and Suggestions for Authors

Comments on the paper submitted to Materials (ISSN 1996-1944)

Title: Numerical study on the effect of coarse aggregate shape during concrete mixing process

Manuscript ID: materials-2934199

General Comments

The paper investigates the influence of coarse aggregate shape on concrete mixing parameters. The paper includes mainly numerical analyses using the particle method. Some experimental analyses were also developed to calibrate certain parameters of the simulations. In summary, the paper addresses an interesting topic with adequate scientific methods. In general, the conclusions are well supported by the presented methods and results.

Based on this, my recommendation is to accept with minor revisions required. I identified only a few places where I suggested that the authors improve the description of the methods or clarify the idea, which can be easily corrected.

Title: The title summarizes the content of the paper well.

Abstract: The Abstract is generally well structured since it provides the motivation of the proposed study, purpose, methods, main results, and conclusions of the paper. Nevertheless, I identified some sentences that need to be revised.

1 – In the sentence “The mixing motion of coarse aggregate and mortar with different shapes was studied using average velocity”. Please, clarify that the different shapes are related to the coarse aggregate.

2 - In the sentences like this one "Spherical coarse aggregate mixes best with mortar, ellipsoidal coarse aggregate mixes worst with mortar." Please, consider improving the flow of ideas by adding adequate connections between the sentences such as "Spherical coarse aggregate mixes best with mortar while ellipsoidal coarse aggregate mixes worst with mortar."

3 - The Introduction is well structured as it provides clearly the motivation for the proposed study, literature review, and main purpose. The research significance is also well highlighted at the end of this section. I suggest only the authors add some Figures in the Introduction to illustrate the content addressed in the paper, such as the effect of different shapes of the coarse aggregate.

4 - In the sentence "Save the mixer model in IGS format for import into EDEM". Please improve the connections between this idea and the previous text. It seems that this sentence is not correctly connected in the paragraph.

5 - Lines 98-100: In the sentence " When the same concrete mix ratio is used, concrete mixed with pebbles has better fluidity but lower strength after hardening, while concrete mixed with crushed stone has poorer fluidity but higher strength after hardening". Please add an explanation to this statement or references to sustain this information.

6 - In the sentence "According to the industry standard [34], when EDEM". This information is not clear...  please, check if reference [34] can be used to support this information. It seems to me that this information is a default set from the software and not from the "industry".

7 - Lines 117-118: "The size of the mortar is set to spherical particles with a radius of 3mm." This set is based on some literature information or previous investigation? Please, clarify.

8 – Line 145-146: remove duplicate information and correct the sentence.

9 - What is the meaning of different colors in Figure 3? Initially, I understood that it would be only two phases (coarse aggregate and mortar). Please, clarify this aspect.

Conclusions

The conclusions are supported by the presented methods and results. Besides, the conclusions are valuable for future investigations in this field.

English writing:

In general, the manuscript is written in good English. Minor revisions on this aspect are sufficient.

Figures and tables.

In general, most Figures were well prepared by the authors.

Comments on the Quality of English Language

In general, the manuscript is written in good English. Minor revisions on this aspect are sufficient.

Reviewer 3 Report

Comments and Suggestions for Authors

The paper “Numerical study on the effect of coarse aggregate shape during concrete mixing process” reports an interesting research work about the evaluation of the influence of the shape and size of the coarse aggregate on the concrete final mechanical properties. In particular, the authors have analysed the behaviour of four different coarse aggregate shape by means of numerical analyses using the discrete element method (DEM). In general, the paper appears well-organized in its different Sections and the topic of the manuscript is of interest for both scientific community and          technologists which work in the concrete industry. Furthermore, the obtained results are discussed in detail in Section 3. For these reasons, it is opinion of this reviewer that the manuscript can be considered for publication in Materials after the following corrections/improvements:

- a significant improvement of the English language is required;

- lines 22-27 consider as reported in 10.1007/978-3-031-37123-3_21 and Longarini N, Crespi P, Zucca M, Giordano N, Silvestro G. “The advantages of fly ash use in concrete structures”. Inzynieria Mineralna 2014, 15(2): pp. 141-145.

- lines 88-90: add the references from which the considered value of the optimal speed for blade starring was determined.

- lines 103-106: Which version of the Solid-Works software was used?

- Table 2: replace “Gpa” with “GPa”;

- improve the quality of Figure 10.

- Check the correct numbering of the Figures, especially after Figure 11.

- a significant improvement of the Section 4 (Conclusion) is suggested highlighting the limits of the results obtained and the further developments of the research work.

Comments on the Quality of English Language

A significant improvement of the English language is required.
